# Transcriptome Data Revealed the circRNA–miRNA–mRNA Regulatory Network during the Proliferation and Differentiation of Myoblasts in Shitou Goose

**DOI:** 10.3390/ani14040576

**Published:** 2024-02-08

**Authors:** Rongqin Huang, Jiahui Chen, Xu Dong, Xiquan Zhang, Wen Luo

**Affiliations:** 1College of Animal Science, South China Agricultural University, Guangzhou 510642, China; 20223139032@stu.scau.edu.cn (R.H.);; 2Guangdong Provincial Key Lab of Agro-Animal Genomics and Molecular Breeding, and Key Laboratory of Chicken Genetics, Breeding and Reproduction, Ministry of Agriculture, Guangzhou 510642, China

**Keywords:** Shitou goose, myogenesis, ceRNA, circRNA

## Abstract

**Simple Summary:**

The Shitou goose, derived from the swan goose (*Anser cygnoides*), is an excellent large goose breed in China and has high economic value. Therefore, it is imperative to delve into the study of its muscle characteristics. Circular RNA (CircRNA), acting as a molecular sponge, binds to miRNA, thereby intricately modulating targeted gene expression and influencing muscle growth and development. To unravel these complexities, we employed transcriptome sequencing to scrutinize the myoblasts and myotubes of Shitou geese, generating comprehensive circRNA and mRNA maps for two distinct developmental stages. In the subsequent analysis, a circRNA–miRNA–mRNA interaction network emerged, delineating a regulatory framework for muscle growth and development. This network is pivotal in understanding the underlying mechanisms of goose myogenesis and contributes new ideas and perspectives concerning the role of non-coding RNAs in the regulation of growth in geese.

**Abstract:**

CircRNA, a recently characterized non-coding RNA (ncRNA) variant, functions as a molecular sponge, exerting regulatory control by binding to microRNA (miRNA) and modulating the expression of downstream proteins, either promoting or inhibiting their expression. Among poultry species, geese hold significant importance, prized by consumers for their delectable taste and rich nutritional content. Despite the prominence of geese, research on the growth and development of goose muscle, particularly the regulatory role of circRNAs in goose muscle formation, remains insufficiently explored. In this study, we constructed comprehensive expression profiles of circRNAs and messenger RNAs (mRNAs) within the myoblasts and myotubes of Shitou geese. We identified a total of 96 differentially expressed circRNAs (DEcircRNAs) and 880 differentially expressed mRNAs (DEmRNAs). Notably, the parental genes of DEcircRNAs and DEmRNAs exhibited enrichment in the Wnt signaling pathway, highlighting its potential impact on the proliferation and differentiation of goose myoblasts. Employing RNAhybrid and miRDB, we identified circRNA-miRNA pairs and mRNA-miRNA pairs that may play a role in regulating myogenic differentiation or muscle growth. Subsequently, utilizing Cytoscape, we constructed a circRNA–miRNA–mRNA interaction network aimed at unraveling the intricate regulatory mechanisms involved in goose muscle growth and development, which comprises 93 circRNAs, 351 miRNAs, and 305 mRNAs. Moreover, the identification of 10 hub genes (*ACTB*, *ACTN1*, *BDNF*, *PDGFRA*, *MYL1*, *EFNA5*, *MYSM1*, *THBS1*, *ITGA8*, and *ELN*) potentially linked to myogenesis, along with the exploration of their circRNA–miRNA–hub gene regulatory axis, was also conducted. These competitive endogenous RNA (ceRNA) regulatory networks elucidate the molecular regulatory mechanisms associated with muscle growth in Shitou geese, providing deeper insights into the reciprocal regulation of circRNA, miRNA, and mRNA in the context of goose muscle formation.

## 1. Introduction

Poultry meat enjoys worldwide popularity owing to its delectable taste and nutritional richness. Among the various poultry varieties, geese stand out as a prominent category. Renowned for its palatable taste and notable richness in protein, fat, and essential amino acids, goose meat boasts distinct nutritional and culinary significance, underlining its considerable economic value [1,2]. The Shitou goose, originating from the swan goose (*Anser cygnoides*), holds the distinction of being the world’s largest body-size goose breed, with its roots tracing back to Raoping County, Guangdong Province, China [3]. Distinguished by its efficiency in grain utilization, substantial body size, enhanced meat production, and rapid growth, this breed presents notable advantages [4]. Despite these merits, research on goose muscle remains limited in comparison to more extensively studied poultry varieties such as chickens and ducks [4,5].

Skeletal muscle holds a pivotal role in the animal organism, performing essential functions such as locomotion, supporting respiration, and maintaining posture [6]. Comprising muscle fibers, muscle bundles, tendons, and muscle membranes, skeletal muscle is intricately structured. Recognized as the fundamental functional unit of skeletal muscle, muscle fibers contribute significantly to its overall functionality [7]. Myogenesis, a complex and orderly process, involves the fusion of myoblasts to form multinucleated myofibrils, meticulously regulated by endogenous factors [8]. In most species, the determination of the number of muscle fibers occurs during the embryonic period. Subsequently, post birth, skeletal muscle growth is confined to the expansion of muscle fiber volume [9,10]. Consequently, the proliferation and differentiation of myoblasts emerge as critical processes influencing the growth and development of skeletal muscle. Furthermore, in livestock and poultry, skeletal muscle stands as a key component contributing to meat consumption by consumers. The yield and quality of meat products exhibit a robust correlation with the growth and development of skeletal muscle [11,12], directly influencing the supply of goose meat and the economic outcomes for breeders. However, diverse goose breeds exhibit varying growth rates and muscle weights. For instance, two native goose breeds in Guangdong Province, the Shitou goose and the Wuzong goose, have attracted considerable attention due to substantial differences in body size. Specifically, at the marketable age, the weight of the Shitou goose can surpass four times that of the Wuzong goose [13,14]. It is noteworthy that recent studies have demonstrated the regulatory role of non-coding RNAs in various stages of skeletal muscle development [15,16].

NcRNA defies the conventional understanding of RNA solely serving as a messenger for protein coding, as it does not participate in such coding processes. Based on their size, ncRNAs can be broadly categorized into various types, with circRNA, miRNA, and long noncoding RNA (lncRNA) emerging as the most extensively researched among them [17,18]. A plethora of studies have illustrated the profound impacts of ncRNAs on organismal development and tissue [17,19]. CircRNA, a type of endogenous ncRNA, is generated through atypical reverse-splicing events. Distinguished by the absence of a 5′-terminal cap and a 3′-terminal poly(A) tail, circRNA forms a closed circular RNA molecule through covalent bonds. This unique structure imparts higher stability to circRNA compared to linear RNA [20,21,22]. The ceRNA hypothesis posits that circRNA contains miRNA response elements (MREs) capable of binding to specific miRNAs, thereby regulating the expression of miRNA target genes. Consequently, circRNA is often referred to as an ‘miRNA sponge’ [23,24,25,26]. Notably, an increasing body of literature underscores the substantial impact of this intricate regulatory network on skeletal muscle growth and development [27,28,29,30,31]. Despite this, the influence of circRNA on goose skeletal muscle still needs to be more adequately explored.

The aim of this study was to investigate the regulatory impact of the ceRNA regulatory network involving circRNA–miRNA–mRNA on the growth and development of skeletal muscle in Shitou goose. Utilizing transcriptome sequencing technology and bioinformatics methods, we analyzed the DEcircRNAs and DEmRNAs during the myogenesis stage of Shitou goose, the largest goose breed in eastern Guangdong, China. Comprehensive analyses of functions and associated regulatory networks were conducted, with the identification of hub genes and subnetworks. This study contributes to the understanding of the molecular regulatory mechanisms underlying myoblast proliferation and differentiation in Shitou goose, providing a theoretical foundation for a more profound comprehension of the molecular regulatory processes governing goose muscle growth and development.

## 2. Materials and Methods

### 2.1. Isolation and Culture of Goose Myoblasts

Fertilized goose eggs were uniformly placed in the same incubator and maintained at 37 °C with 70% humidity. Following 15 days of incubation, the thigh muscles from 10 Shitou goose embryos were meticulously isolated and sectioned using sterile surgical instruments. The minced muscle tissue underwent digestion with 10 mL of 0.25% trypsin for 20 min in a constant temperature incubator, and digestion was halted with 10 mL of DMEM medium containing 20% fetal bovine serum (FBS) (Gibco, Grand Island, NE, USA). Subsequently, the digested suspension was transferred to a 50 mL centrifuge tube and centrifuged at 1200× *g* for 5 min, and the supernatant was discarded. Resuspension of the cells took place in DMEM medium containing 15% FBS, and they were evenly seeded in a petri dish. This continuous plating method aimed to enrich myoblasts while eliminating fibroblasts. The mixed primary myoblasts were then seeded across six culture dishes. Of these, three cultures were designated for the collection of growth myoblasts (GM), while the remaining three cultures underwent induction for differentiation into myoblasts using 2% horse serum (HS) (Biosharp, Hefei, China) in lieu of the 20% fetal bovine serum and were used to collect differentiated myotubes (DMs) [11].

### 2.2. RNA Isolation, Library Construction, and Sequencing

Once the myoblasts achieved a proliferation density of 70%, they underwent three washes with PBS and were directly added to the Trizol reagent (Invitrogen, Carlsbad, CA, USA). For the differentiated myoblasts, when their density reached 80%, they were cultured in DMEM containing 2% HS. Following three days of differentiation, the samples were collected using Trizol. Subsequently, total RNA extraction was conducted according to the manufacturer’s instructions. The specific steps were as follows: after washing the cells three times with PBS, 1 mL of Trizol reagent was added directly to the culture dish and placed on ice for 5 min to facilitate complete lysis. After repeated aspiration, the solution was transferred into a 1.5 mL RNase-free centrifuge tube, and 200 μL of chloroform was added. The solution was thoroughly mixed and left on ice for 5 min before centrifugation at 4 °C, 1200× *g* for 15 min. In a new RNase-free centrifuge tube, 500 μL isopropanol was added, and the supernatant obtained by means of centrifugation was aspirated. The solution was shaken up and down again to mix it well and centrifuged at 4 °C, 1200× *g* for 10 min. The supernatant was discarded, 1 mL of 75% ethanol was added, and the tube was centrifuged at 4 °C, 7500× *g* for 5 min. The supernatant was discarded, and the tube was left on ice for 5 min to dry the tube wall. Finally, 30 μL DEPC solution was added to dissolve the RNA. The Nanodrop ND-1000 (NanoDrop, Wilmington, DE, USA) was utilized to quantify the amount and purity of RNA in each sample. We employed the Agilent 2100 system to assess RNA integrity, where the RNA integrity number (RIN) of ≥7.0 is commonly regarded as optimal for subsequent high-throughput applications. In our experiment, we strictly adhered to the criterion of utilizing RNA with RIN > 7.0 to ensure the integrity of the RNA samples [32].

Ribosomal RNA was eliminated by utilizing 5 μg of total RNA in accordance with the instructions provided by the Ribo-Zero™ rRNA Removal Kit (Illumina, San Diego, CA, USA). Following the removal of ribosomal RNAs, the remaining RNAs were fragmented into small pieces using divalent cations under high temperatures. A volume of 8.5 μL cleaved RNA fragments was reverse-transcribed with random primers to generate cDNA, subsequently used to synthesize U-labeled second-stranded DNAs employing *E. coli* DNA polymerase I, RNase H, and dUTP. Blunt ends were introduced to each strand by adding an A-base, preparing them for ligation to the indexed adapters. Each adapter, featuring a T-base overhang, facilitated ligation to the A-tailed fragmented DNA. Fragments were ligated with single- or dual-index adapters, and size selection was executed using AMPureXP beads. The U-labeled second-stranded DNAs underwent treatment with the heat-labile UDG enzyme.

Ligated products were then amplified via PCR under the following conditions: initial denaturation at 95 °C for 3 min; 8 cycles of denaturation at 98 °C for 15 s, annealing at 60 °C for 15 s, and extension at 72 °C for 30 s; and a final extension at 72 °C for 5 min. The resulting cDNA library exhibited an average insert size of 300 bp (±50 bp). Finally, double-end sequencing was conducted on an Illumina NovaseqTM 6000 (LC Bio, Hangzhou, China), following the protocol recommended by the supplier.

### 2.3. Read Mapping and Transcriptome Assembly

Cutadapt [33] was employed to eliminate reads containing adaptor contamination, low-quality bases, and undetermined bases. Subsequently, sequence quality was assessed using FastQC (http://www.bioinformatics.babraham.ac.uk/projects/fastqc/ accessed on 7 May 2021). Read mapping to the species genome was carried out utilizing both Bowtie2 [34] and Hisat2 [35]. For any remaining reads (unmapped reads), a supplementary mapping step to the genome was performed using tophat-fusion [36].

### 2.4. Identification of circRNAs

CIRCExplorer2 [37,38] and CIRI [39] were employed for de novo assembly of the mapped reads into circular RNAs initially. Subsequently, back-splicing reads were identified within the unmapped reads using tophat-fusion. This process ensured the generation of unique circular RNAs for all samples.

### 2.5. Differential Expression Analysis of circRNAs and mRNAs

To identify DEcircRNAs and DEmRNAs between myoblasts and myotubes, edgeR was employed for differential expression analysis. A threshold of |log2FC| ≥ 1 and *p*-value < 0.05 was applied for the analysis.

### 2.6. Prediction of Targeting Relationship

To identify the target miRNAs of DEcircRNAs, we utilized RNAhybrid (https://bibiserv.cebitec.uni-bielefeld.de/rnahybrid accessed on 16 October 2023) to predict the interacting miRNAs. Additionally, we used miRDB (https://mirdb.org/ accessed on 16 October 2023) to predict the target miRNA that interacts with mRNA, with a threshold set to ‘Target Score > 90′. The final miRNA (FmiRNA) was obtained by intersecting the two predicted miRNAs.

### 2.7. Functional Enrichment Analysis

Gene Ontology (GO) and Kyoto Encyclopedia of Genes and Genomes (KEGG) enrichment analysis along with the visualization of DEcircRNA host genes and DEmRNAs were conducted using the R package clusterProfiler.

For a deeper insight into gene functions, gene set enrichment analysis (GSEA) and its visualization for mRNAs were carried out using the R packages clusterProfiler (version 4.10.0), enrichplot (version 1.22.0), and ggstatsplot (version 0.12.1). The clusterProfiler package’s gseGO and gseKEGG functions were utilized for the GO and KEGG enrichment analyses of gene sets, respectively. Visualization of the results from the GSEA was carried out using Enrichplot and ggstatsplot. The set threshold was established at |NES| > 1, NOM *p*-value < 0.05, and FDR (padj) < 0.25.

### 2.8. Establishment of the circRNA–miRNA–mRNA/hub Gene Network

The circRNA–miRNA and mRNA–miRNA pairs were integrated to eliminate nodes that could not contribute to the formation of the circRNA–miRNA–mRNA axis. The resulting circRNA–miRNA–mRNA interaction network was visually represented using Cytoscape (version 3.10.0).

After identifying 10 hub genes, we integrated them into the original circRNA–miRNA–mRNA network to explore the relationships between circRNAs, miRNAs, and hub genes. Subsequently, relevant circRNAs and mRNAs were extracted, and the circRNA–miRNA–hub gene subnetwork was reconstructed using Cytoscape.

### 2.9. Protein–Protein Interaction (PPI) Network Analysis

The Search Tool for the Retrieval of Interacting Genes/Proteins (STRING) database (https://string-db.org/ accessed on 19 October 2023) was utilized to discern the associations among mRNAs. The interaction network of PPI was analyzed using the CytoHubba application in Cytoscape. The top 10 hub genes in the interaction network were determined based on degree.

## 3. Results

### 3.1. Analysis of DEcircRNAs in the Proliferation and Differentiation Stages of Myoblasts in Shitou Goose

To discern variations in the proliferation and differentiation of myoblasts, transcriptome sequencing of cells at GM and DM phases was conducted using the Illumina HiSeq platform. The sequencing quality across all samples remained consistently high, as confirmed by the box plot (Figure 1A). A total of 5612 circRNAs were identified and distributed across different chromosomes during the transcriptome sequencing process. The classification results indicated that 81.27% of circRNAs were located in exons, 15.41% were located in introns, and 3.32% were located in intergenic regions (Figure 1B, Appendix A). In total, 96 DEcircRNAs were identified in the proliferation and differentiation stages of myoblasts, applying a threshold of |log2FC| ≥ 1 and a *p*-value < 0.05. Among these, 76 were upregulated, and 20 were downregulated (Figure 1C,D, Appendix A).

After mapping the DEcircRNAs to their parental genes, the subsequent GO and KEGG enrichment analysis revealed significant enrichment. The results of the enrichment analysis showed that the genes in the process of myoblast proliferation and differentiation were predominantly enriched in metal ion binding, cation binding, endosome membranes, endosomes, cytoplasmic vesicle membranes, vesicle membranes, phagosomes, and Wnt signaling pathway (Figure 1E,F).

### 3.2. Analysis of DEmRNAs between Myoblasts and Myotubes in Shitou Goose

Following the circRNA analysis, we identified 880 DEmRNAs from a pool of 20,208 mRNAs (Appendix A), with 536 mRNAs exhibiting upregulation and 344 mRNAs showing downregulation (Figure 2A,B, Appendix A). The subsequent GO and KEGG analyses aimed to unveil critical genes and pathways associated with goose muscle development. The analysis of GO revealed enrichment in biological pathways associated with cell development and differentiation, including cell differentiation, cellular developmental processes, animal organ development, and tissue development. The KEGG results also pointed towards the involvement of these genes in related biological signaling pathways, such as the MAPK signaling pathway, the calcium signaling pathway, and focal adhesion (Figure 2C,D). Subsequently, we ranked mRNAs using log2FC and conducted GSEA to elucidate gene functions. The GSEA results validated the earlier enrichment content (Figure 2E–G).

### 3.3. Construction of circRNA–miRNA Pairs and miRNA–mRNA Pairs

Our objective was to unveil miRNAs interacting with DEcircRNAs. The mature sequences of miRNAs were obtained from miRbase (https://mirbase.org accessed on 16 October 2023). Utilizing RNAhybrid prediction, we identified 692 target miRNAs, establishing a total of 4076 circRNA–miRNA interaction pairs (Appendix A). These pairs were determined based on the perfect pairing of the second to eighth nucleotides of DEcircRNAs at the 5′ end of the miRNA sequence. Simultaneously, miRDB was employed to pinpoint miRNAs interacting with DEmRNAs. We acquired 681 miRNAs, resulting in 2602 mRNA–miRNA pairs (Appendix A) based on a target score > 90. To enhance prediction accuracy, we intersected the two sets of predicted target miRNAs, yielding 351 FmiRNAs (Figure 3A, Appendix A).

Studies have elucidated that miR-204 can impede the proliferation and differentiation of chicken skeletal muscle satellite cells [40]. Moreover, miR-204 orchestrates myogenesis and skeletal muscle regeneration in mice by binding to specific genes [41]. Our prediction highlights that miR-204 forms circRNA–miRNA pairs with circRNA1188, circRNA4337, and ciRNA781 (Figure 3B). Additionally, miR-204 establishes mRNA–miRNA pairs with *TM6SF1*, *BICC1*, *MGAT3*, *PIK3IP1*, *RASA3* and *EFNA5*. Furthermore, we have identified several pertinent genes, such as *myosin light polypeptide 1* (*MYL1*) and miR-3532-3p, *EFNA5* and miR-1575, miR-10585-5p, miR-204, miR-211, and miR-12280-3p, forming mRNA–miRNA pairs (Appendix A). These genes have been demonstrated to exert regulatory effects on livestock and poultry myogenesis [42,43,44,45].

### 3.4. Construction of the circRNA–miRNA–mRNA Interaction Network

Studies have emphasized the capacity of circRNA to act as a ceRNA, functioning as a molecular sponge to competitively bind miRNA and regulate the expression of miRNA target genes [46,47]. Our objective was to identify the ceRNA interaction network linked to myogenesis. By filtering circRNAs and mRNAs regulated by the same FmiRNA expression, we integrated circRNA-miRNA pairs and miRNA-mRNA pairs using Cytoscape software, thereby constructing a preliminary circRNA–miRNA–mRNA interaction network consisting of 749 nodes and 2867 edges (Figure 4, Appendix A).

### 3.5. Establishment of the PPI Network and Identification of Hub Genes

To further identify the interaction network among mRNAs and enhance our understanding of gene–gene interactions, we utilized the STRING database and Cytoscape. The analysis revealed 125 genes with interaction relationships, forming a PPI interaction network of 125 nodes and 192 edges (Figure 5A). Additionally, we utilized the cytoHubba plug-in within Cytoscape software to rank mRNAs based on their Degree. The Degree value indicates the gene’s importance, with higher values indicating greater significance. The top 10 genes, identified as hub genes, are crucial nodes in the regulatory network, including *ACTB* (*actin beta*), *ACTN1* (*actinin alpha 1*), *BDNF* (*brain-derived neurotrophic factor*), *PDGFRA* (*platelet-derived growth factor receptor alpha*), *MYL1* (*myosin light chain 1*), *EFNA5* (*ephrin-A5*), *MYSM1* (*Myb-like*, *SWIRM*, and *MPN domains 1*), *THBS1* (*thrombospondin 1*), *ITGA8* (*integrin subunit alpha 8*), and *ELN* (*elastin*) (Figure 5B, Table 1).

### 3.6. Reconstructing the circRNA–miRNA–Hub Gene Interaction Network

Incorporating the 10 hub genes from the initial circRNA–miRNA–mRNA interaction network, we examined associated circRNAs and miRNAs, by constructing a circRNA–miRNA–hub gene interaction network. This network, further categorized as a circRNA–miRNA–mRNA subnetwork, comprised 44 circRNAs, 30 miRNAs, 10 mRNAs, and 156 edges (Figure 6). In total, 126 circRNA–miRNA–mRNA regulatory axes were identified within the subnetwork (Appendix A).

## 4. Discussion

As an endogenous ncRNA, circRNA is produced through the reverse splicing of mRNA precursors (pre-mRNA). It is a recent addition to the family of ncRNAs, following miRNA and lncRNA, and has great research potential. CircRNA possesses features such as high stability, universality, and tissue cell specificity [20]. Research has revealed that circRNA usually contains abundant miRNA binding sites and functions as a ceRNA. It competes for the binding of shared miRNAs through partial complementary sequences, also referred to as MREs, which leads to the degeneration of target genes or the inhibition of translation at the post-transcriptional level, thereby controlling biological processes [48,49,50,51,52]. In recent years, studies have shown that the ceRNA network has a significant impact on myogenesis. For instance, Chen et al. demonstrated that circMYBPC1 effectively binds with miR-23a, reducing its inhibition of *myosin heavy chain* (*MYHC*), and promotes myoblast differentiation [53]. Similarly, Shen et al. found that muscle-specific circGPD2 releases the inhibitory effect of miR-203a on *c-JUN* and *MEF2C* [54]. The objective of our study was to investigate the existence of a ceRNA network during the process of goose muscle formation. To achieve this, we utilized myoblasts from the Shitou goose as research materials to identify DEcircRNAs and DEmRNAs during the GM and DM periods. We then employed various bioinformatics prediction methods to identify miRNAs that interact with each other, resulting in the formation of a circRNA–miRNA–mRNA interaction network that may play a role in regulating goose myogenesis. However, compared to model animals such as chickens and mice, research on the regulation of circRNA in goose skeletal muscle is still at an early stage. Currently, only a limited number of circRNA and miRNA identifications have been made in geese. The existing circRNA and miRNA databases are predominantly focused on chickens, mice, and other model animals. Consequently, there is currently no database for geese that incorporates tissue-specific expression patterns for circRNAs and miRNAs. This limitation may lead to the omission of certain circRNAs and miRNAs with spatiotemporal expression patterns or those that are unannotated.

Livestock and poultry have various signaling pathways that regulate muscle growth, such as the P53 signaling pathway [55], the Notch signaling pathway [56], the mTOR signaling pathway [57], and the p38 MAPK signaling pathways [58]. In this study, we conducted enrichment analysis on the parental genes of DEcircRNAs and DEmRNAs, revealing enrichment in the Wnt signaling pathway. The Wnt protein functions as a cell signaling molecule, primarily stimulating cell proliferation, differentiation, and migration [59]. Numerous studies have demonstrated the significant role of the Wnt signaling pathway in both development and disease [60,61,62]. Additionally, some researchers have identified a close relationship between the Wnt signaling pathway and myogenesis. For instance, knockout mice of core components of the Wnt signal transduction pathway typically exhibit inadequate muscle development and high lethality [63]. The *miR-136-5p/FZD4* axis regulates myogenesis and muscle regeneration through the Wnt signaling pathway [64]. Our results also support this notion, highlighting the crucial role of Wnt signaling in Shitou goose myogenesis.

The formation of skeletal muscle in livestock and poultry is a complex biological process. It involves coding genes, signaling pathways, and ncRNAs [65,66]. *Myostatin* (*MSTN*), a member of the transforming growth factor β superfamily, regulates skeletal muscle growth and development through autocrine or paracrine signal transduction [54]. H19 lncRNA increases muscle insulin sensitivity by activating AMPK [67]. In this study, 10 hub genes were extracted using the PPI interaction network. One of these genes, *ACTB*, encodes β-actin, which is associated with cell shape, migration, proliferation, and gene expression, and can therefore affect organ development [68]. Another related gene for cytoskeletal actin is *ACTN1*, which is commonly expressed in most tissues and cell types. In muscle cells, actin connects adjacent muscle segments and coordinates muscle contraction [69,70]. *MYL1* plays a crucial role in embryonic skeletal muscle development, and its mutation can result in the loss or malnutrition of type II muscle fibers. It is a key gene in ensuring skeletal muscle integrity [42,43]. *EFNA5* is a member of the ephrin-A family of protein genes. Previous studies have shown that this gene can directly or indirectly affect the growth of motor neurons adjacent to it [71]. Researchers have found that interactions between motor neurons and muscle stem cells contribute to the establishment of functional muscle during development and regeneration [72]. Additionally, He et al. discovered a negative correlation between the expression level of *EFNA5* and the weight of chicken pectoral muscles. *EFNA5* has the potential to regulate muscle development [73]. *ITGA8* mediates various cellular processes, including cytoskeleton rearrangement and cell adhesion. Studies have shown that this gene helps regulate cell proliferation and the apoptosis of glomerular cells [74]. Here, we found that it is differentially expressed between myoblasts and myotubes and forms mRNA-miRNA pairs with multiple miRNAs, indicating that its regulatory influence on other cells may also be established in myoblasts.

Although we constructed a ceRNA regulatory network that regulates the proliferation and differentiation of goose myoblasts and screened potential key genes and the related circRNA–miRNA–hub gene regulatory axis, there are still some limitations. Firstly, this experiment only used the transcriptome data of the proliferation and differentiation stages of myoblasts of the Shitou goose as the data source. Additional prospective studies, involving multiple goose breeds and a larger sample size, are necessary to validate our findings. Furthermore, while we have implemented rigorous standards in our bioinformatics prediction process to enhance the dependability of our regulatory network, the potential for false positives cannot be completely eliminated. Further experimental studies are required to confirm the potential biological regulatory mechanisms of these ceRNAs in myogenesis.

## 5. Conclusions

In this study, we leveraged transcriptome data and employed bioinformatics analysis methods to assemble the expression profiles of circRNAs and mRNAs in the proliferation and differentiation of myoblasts in Shitou goose. We identified 96 DEcircRNAs and 880 DEmRNAs during the two periods. The functional annotation results indicate that the parental genes of DEcircRNAs and DEmRNAs were enriched in the Wnt signaling pathway, suggesting a crucial role of the Wnt signaling pathway in regulating the proliferation and differentiation of adult myoblasts. Additionally, we constructed a circRNA–miRNA–mRNA ceRNA regulatory network that could potentially participate in myogenesis. We identified 10 highly related hub genes (*ACTB*, *ACTN1*, *BDNF*, *PDGFRA*, *MYL1*, *EFNA5*, *MYSM1*, *THBS1*, *ITGA8*, and *ELN*) involved in goose muscle growth and development, along with their related circRNA–miRNA–hub gene regulatory axis. In conclusion, our analysis of transcriptome data from the myoblasts and myotubes of the Shitou goose enabled us to construct a comprehensive ceRNA regulatory network. This elucidated how circRNA regulates the mechanism underlying myoblast proliferation and differentiation. These findings establish a robust foundation for comprehending the mechanism and regulatory network of goose muscle formation, contributing valuable insights to further research and development in this field.

## Figures and Tables

**Figure 1 animals-14-00576-f001:**
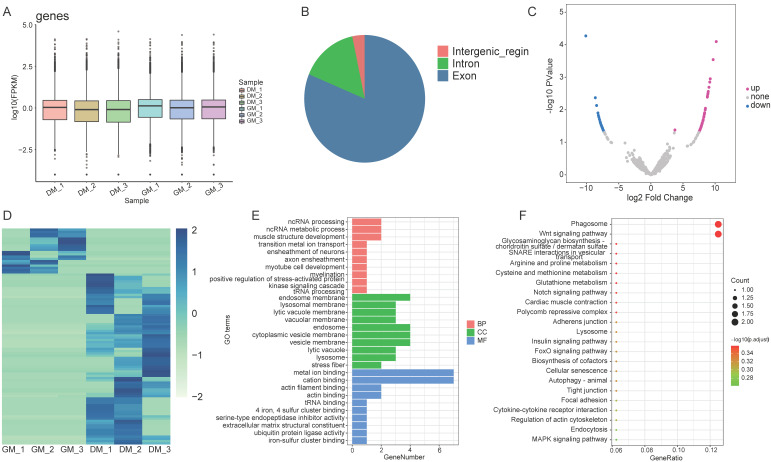
Analysis of circRNAs in myoblasts and myotubes of the Shitou goose. (**A**) Transcript expression levels for each sample showcase consistently high sequencing quality, confirmed by a box plot. (**B**) CircRNAs were identified and classified based on their gene source, revealing their distribution across exons, introns, and intergenic regions. (**C**,**D**) Volcano maps (**C**) and heat maps (**D**) visually represent the DEcircRNAs in myoblasts and myotubes. Upregulated circRNAs are marked in red, while downregulated ones are shown in blue. (**E**,**F**) GO enrichment (**E**) and KEGG pathway (**F**) analysis of DEcircRNAs parental genes in myoblasts and myotubes.

**Figure 2 animals-14-00576-f002:**
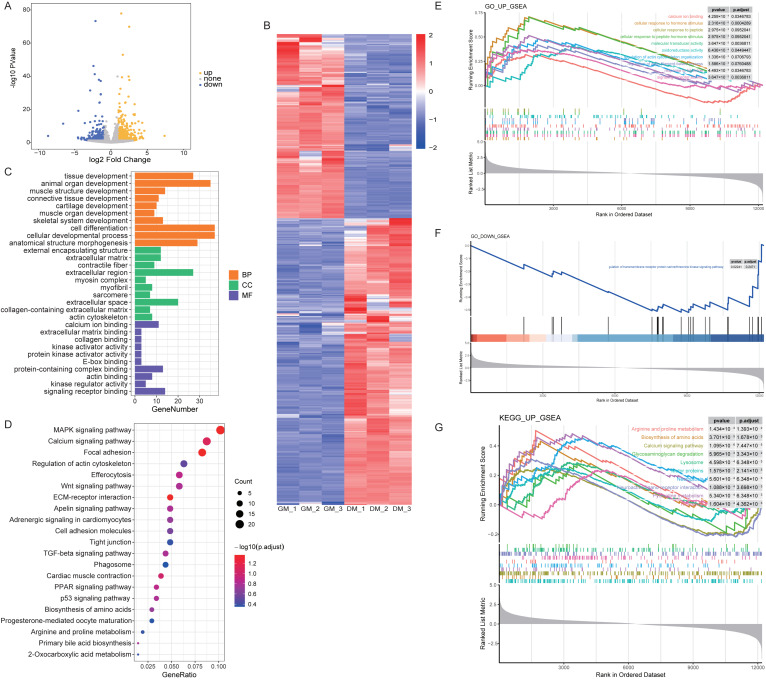
Analysis of mRNAs between myoblasts and myotubes in Shitou goose. (**A**,**B**) Volcano plots (**A**) and heat maps (**B**) visually representing the DEmRNAs in myoblasts and myotubes. The upregulated mRNAs are depicted in yellow, while the downregulated ones are depicted in blue. (**C**,**D**) GO enrichment analysis (**C**) and KEGG pathway enrichment analysis (**D**) of DEmRNAs in myoblasts and myotubes. (**E**,**F**) GSEA analysis revealed significantly enriched GO terms for upregulated (**E**) and downregulated (**F**) mRNAs. (**G**) The GSEA analysis highlighted significantly enriched upregulated KEGG pathways.

**Figure 3 animals-14-00576-f003:**
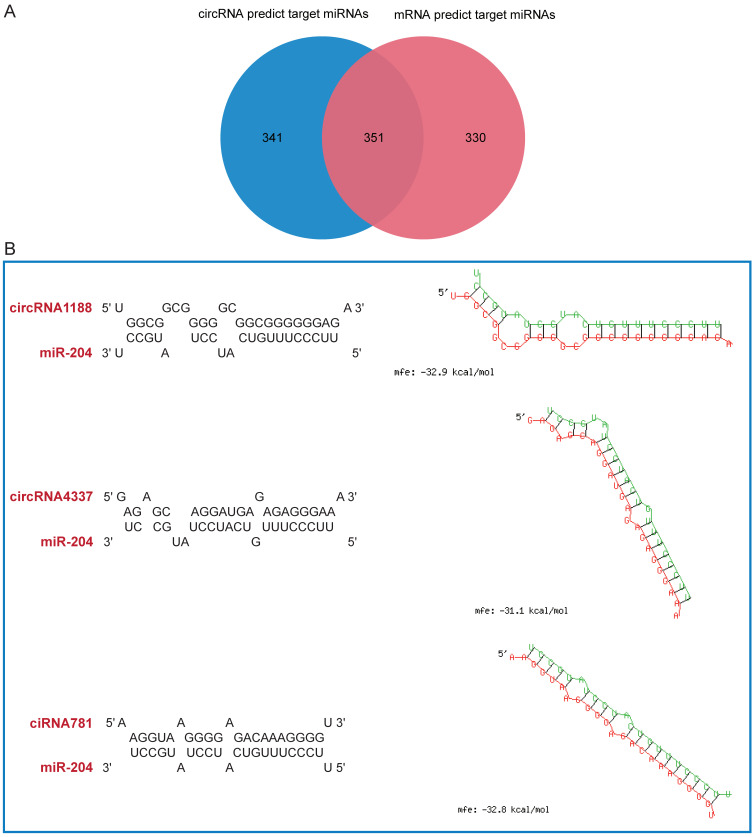
Construction of circRNA–miRNA pairs and mRNA–miRNA pairs. (**A**) Venn diagram illustrating target miRNAs. The blue circle denotes the miRNAs predicted by circRNA, the red circle represents miRNAs predicted by mRNA, and the overlap signifies FmiRNAs. (**B**) Schematic depiction of binding sites between miR-204 and circRNA1188, circRNA4337, and ciRNA781. Green lines signify miRNA, and red lines signify circRNA.

**Figure 4 animals-14-00576-f004:**
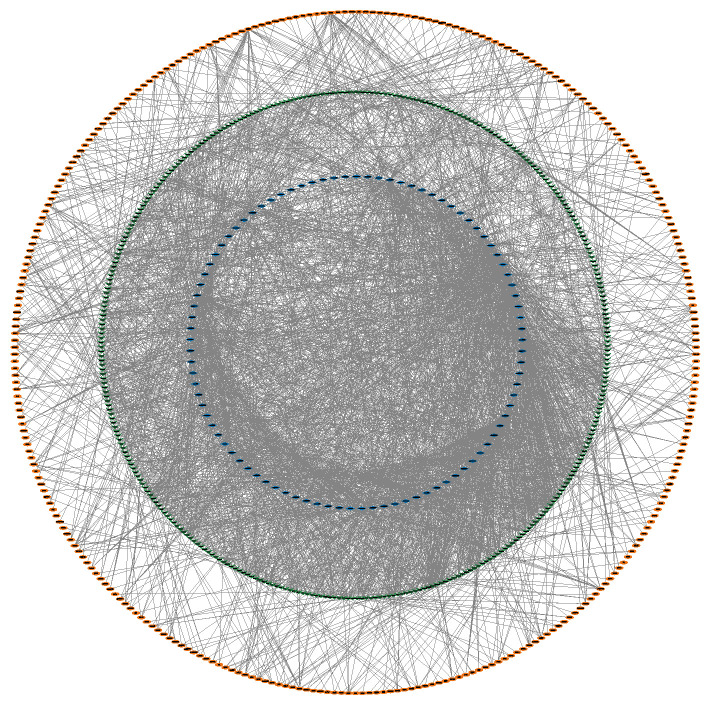
circRNA–miRNA–mRNA interaction network diagram. The blue diamond represents circRNAs, the green triangle represents miRNAs, and the yellow oval represents mRNAs.

**Figure 5 animals-14-00576-f005:**
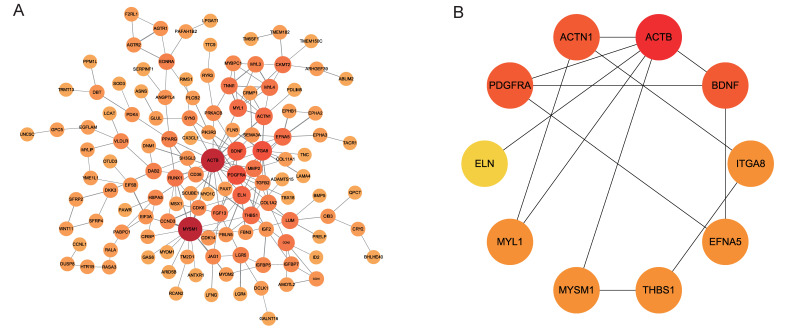
PPI interaction network. (**A**) PPI interaction network diagram of mRNAs. (**B**) Hub genes interaction network diagram. The size and color of the circles correspond to the degree score.

**Figure 6 animals-14-00576-f006:**
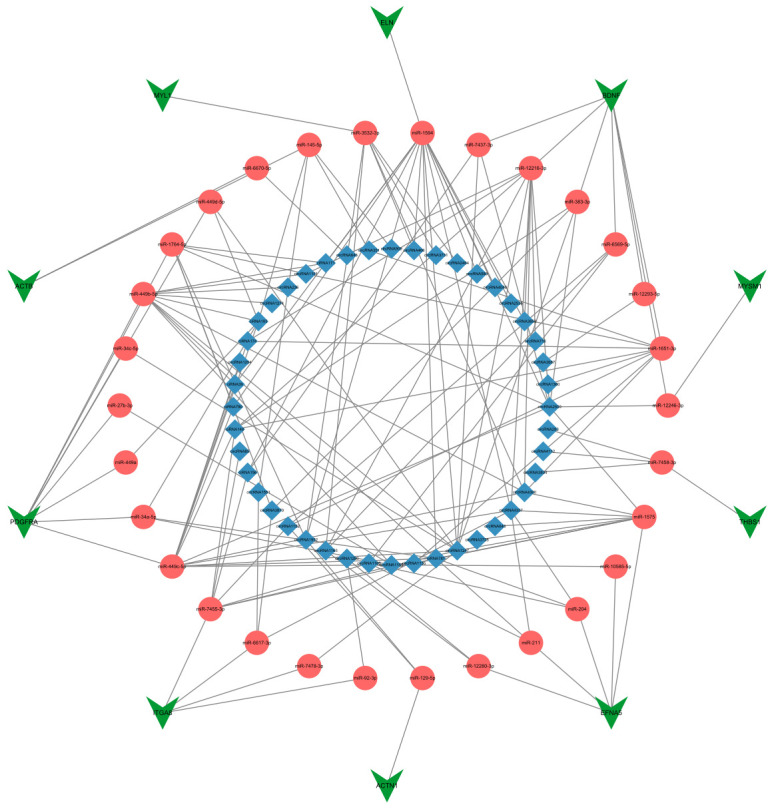
The circRNA–miRNA–hub gene interaction network diagram. The blue diamond represents circRNAs, the red circle represents miRNAs, and the green triangle represents mRNAs.

**Table 1 animals-14-00576-t001:** Ten hub genes obtained by degree.

Name	Score	Gene Description
*ACTB*	6	*actin beta*
*ACTN1*	3	*actinin alpha 1*
*BDNF*	3	*brain-derived neurotrophic factor*
*PDGFRA*	3	*platelet-derived growth factor receptor alpha*
*MYL1*	2	*myosin light chain 1*
*EFNA5*	2	*ephrin-A5*
*MYSM1*	2	*Myb-like, SWIRM, and MPN domains 1*
*THBS1*	2	*thrombospondin 1*
*ITGA8*	2	*integrin subunit alpha 8*
*ELN*	1	*elastin*

## Data Availability

The datasets supporting the conclusions of this article are available in GEO: https://www.ncbi.nlm.nih.gov/geo/query/acc.cgi?acc=GSE213147, accessed on 15 September 2022.

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
