# Peer review of "Transcriptome Data Revealed the circRNA–miRNA–mRNA Regulatory Network during the Proliferation and Differentiation of Myoblasts in Shitou Goose"

_animals, 2024, doi:10.3390/ani14040576_

Round 1

Reviewer 1 Report

Comments and Suggestions for Authors

CircRNA, characterized by its closed-loop structure, is a subtype of ncRNA that plays a crucial regulatory role in the growth and development of livestock and poultry. It holds significance in sustaining vital life processes, fostering development, and adapting to environmental shifts. This study explores the transcriptome of Shitou goose myoblasts during both proliferative and differentiated stages. The analysis delineates circRNA-miRNA pairs and mRNA-miRNA pairs implicated in the potential regulation of myoblast differentiation. Additionally, an intricate circRNA-miRNA-mRNA interaction network governing goose myogenesis is constructed. While the research findings largely corroborate the main points presented in this manuscript, there are still areas requiring refinement at this juncture. The following specific suggestions are provided to enhance the manuscript:

Materials and Methods

(line: 129-131): Why is the threshold set to Target Score > 90? Can you explain this?

The methods described in sections ' 2.8. Establishment of the circRNA-miRNA-mRNA network ' and ' 2.10. Reconstruction of the circRNA-miRNA hubgene network ' are repeated and it is recommended that they be merged into the same section.

Results

(line: 249): The function of the top ten hub genes and their effects on muscle growth and development will be discussed in more detail in the Discussion section.

Reviewer 2 Report

Comments and Suggestions for Authors

Dear Authors,

I have a very substantive remark to your article.

The article makes a strange impression.  On the one hand, it presents extensive material for analysis, on the other hand, this analysis has not been done.  For this large, complex, very expensive and, I think, conscientious work to make sense, a comparison between myogenesis in Anser cygnoides and its specialized offspring, the Shitou goose, which exhibits growth acceleration and rapid muscle mass gain, should be made. At the very least, you need to compare Shitou goose with geese of other breeds. The study subject itself (Shitou goose) cannot be considered as a model for studying myogenesis without comparative bioinformatic analysis with the "norm" for geese. Otherwise, it cannot be established which molecular players are involved basally and which have altered expression in response to artificial selection.   At the moment, the paper looks like a set of facts that should precede a full-fledged publication. This is borne out by the helplessly written "Discussion" chapter, since there is essentially nothing much to discuss at this point.

Small remarks:

1. Simple Summary: Shitou goose, a prominent large goose species indigenous to China, holds significant economic importance, warranting in-depth investigations into its muscle physiology.

- breed, derived from the swan goose (Anser cygnoides)

2. Introduction: Despite these merits, research on goose muscle remains limited in compari- son to more extensively studied poultry varieties such as chickens and ducks. Skeletal muscle holds a pivotal role in the organism, performing essential functions such as exercise, supporting respiration, and maintaining posture [3]

            - references are needed

            - locomotion?

3. Materials and Methods, 2.1. Isolation and Culture of Goose Myoblasts The thigh muscles from ten Shitou geese, hatched over a period of fifteen days, were meticulously dissected using sterile surgical tweezers.

            -  what do you mean? Are the geese 15 days old after hatching?

            - If the methodology for obtaining the cells is not original, then a reference is needed

4. Results. 3.3. Construct circRNA-miRNA pairs and miRNA-mRNA pairs. These pairs were determined based on the perfect pairing of the second and eighth nucleotides of DEcircRNAs at the 5' end of the miRNA sequence.

- Two nucleotides are not enough here. Did you mean a match at the level of the second to eighth nucleotide at the 5'-end of miRNAs?

5. To enhance prediction accuracy, we intersected the two sets of predicted target miRNAs, yielding 351 FmiRNAs (Figure 3A, Table S7).

            - what does "fmiRNAs" mean?

6. Discussion.

            - The "Discussion" chapter is largely the same as the "Introduction" chapter.

7. LncRNA increases muscle insulin sensitivity by activating AMPK [44]

- Are we talking about a specific LncRNA? In this case, it needs to be named. Is there any reason to write about LncRNAs at all if they are not the focus of this article?

8. While our study has meticulously adhered to heightened standards to fortify the reliability of our circRNA and regulatory network, it is essential to acknowledge certain limitations.

            - This needs to be said in more detail so as not to mislead the readers.

Reviewer 3 Report

Comments and Suggestions for Authors

Animals-2822545-peer-review-Report-1

This paper investigated myoblasts and myotubes in Shitou geese, the researchers meticulously compiled detailed expression profiles for circRNAs and messenger RNAs (mRNAs). Uncovering 96 differentially expressed circRNAs (DEcircRNAs) and differentially expressed mRNAs (DEmRNAs), this study lays the groundwork for subsequent research with potential applications in breeding programs and therapeutic interventions aimed at enhancing goose muscle development. The paper is well-written and has a significant contribution to knowledge in this area. However, I have taken time to point out areas that require attention to improve the paper. Addressing these would contribute to a more detailed, comprehensible, and impactful paper.

Abstract

The abstract could be improved by addressing or incorporating the following:

a)     The abstract mentions the identification of differentially expressed circRNAs and mRNAs, but it lacks specific findings or insights gained from this analysis. Providing examples or key outcomes could enhance the impact of the abstract.

b)    The abstract does not include information on the methodologies employed to identify differentially expressed circRNAs and mRNAs. Including details about the experimental design, sequencing platforms, and statistical analyses would enhance the credibility of the study.

c)     While the abstract mentions the construction of circRNA-miRNA-mRNA interaction networks, it lacks information on the functional significance of these networks. Providing insights into the potential biological roles of identified circRNAs and their impact on muscle growth would strengthen the abstract.

d)    Integrating a brief overview of the existing knowledge and positioning the study within the broader field would enhance its significance.

e)     It should address potential applications in goose breeding or biotechnology to add practical relevance.

Introduction

The introductory section of the paper provides a good overview of the research focus, there are areas that could be improved:

a)     The transition between discussing the economic importance of goose meat and the introduction of non-coding RNAs (ncRNAs) could be smoother. Consider providing a clearer link between the economic significance of goose meat and the need for understanding molecular regulatory mechanisms.

b)    While it mentions the limited research on goose muscle compared to chickens and ducks, the paper could explicitly state the specific gaps or knowledge deficiencies in the current literature that the study aims to address.

c)     The paper mentions the importance of skeletal muscle in poultry, but it could elaborate more on the unique features or challenges related to goose skeletal muscle, emphasizing why it warrants specific attention.

d)    While the paper introduces the concept of ncRNA and circRNA, a more comprehensive background on the roles and functions of ncRNA, especially circRNA, would be helpful for readers less familiar with the topic.

e)     The introduction could benefit from a succinct conclusion summarizing the identified gaps in knowledge, the significance of the study, and the specific aims, providing a smooth transition to the methods section.

f)      Lines 60/61: Paraphrasing is required to improve understanding. Suggestion: Despite this, the influence of circRNA on goose skeletal muscle still needs to be more adequately explored.

Materials and Methods

This section lacks some essential ingredients, and these must be addressed.

a)     There is no information on ethical approval; it could be more prominently placed here, if available. Include the date of approval.

b)    The authors have jumped into the description of the RNA extraction without giving a breakdown of the experimental design. It is important to give a clear description of selection, breeding, dietary treatments (if any) and housing. If already presented elsewhere, give the reference.

c)     The authors need to provide more details about the isolation and culture of goose myoblasts by paying attention to the following:

       i.          Specify the time frame for the hatching period of the Shitou geese. If the hatching occurred over a specific period, provide that information for clarity.

     ii.          Elaborate on the concentration of trypsin used and the volume applied during the digestion process. Providing this information enhances the reproducibility of the experiment.

   iii.          Specify the volume of DMEM medium supplemented with 20% foetal bovine serum used to terminate the digestion process. This information is essential for the replication of the experiment.

   iv.          Specify the parameters used for centrifugation, such as the rotor type, time, and speed at 1200 g for 5 minutes.

     v.          Provide more details on the induction process for differentiation, particularly the rationale behind using 2% horse serum instead of 20% foetal bovine serum. Explain the choice of Biosharp (Hefei, China) for horse serum.

RNA Isolation, Library Construction and Sequencing:

a)     The authors need to give a brief account of how RNA was extracted following the manufacturer’s instructions. Specify the kit used as well. This is essential for reproducibility by other interested researchers.

b)    Offer brief information on why the RIN value is significant for assessing RNA integrity. This will help readers understand the quality control measures applied.

c)     Provide more information on the cDNA synthesis process, especially the amounts of RNA used for reverse transcription.

d)    Specify the primer sequences used and any other details regarding the PCR amplification, such as the number of cycles.

Following RNA Isolation, Library Construction, and Sequencing, subsequent paragraphs detail read mapping, transcriptome assembly, circRNA identification, differential expression analysis, targeting prediction, functional enrichment, circRNA–miRNA–mRNA network establishment, PPI network analysis, and circRNA–miRNA–hubgene network reconstruction methods. Here are a few suggestions for adjustments:

a)     In the "Functional Enrichment Analysis" section, provide more details on how GSEA results were visualized using clusterProfiler, enrichplot, and ggstatsplot. Explain the specific functions or features utilized.

b)    In the "Establishment of the circRNA–miRNA–mRNA network" section, briefly explain why Cytoscape (version 3.10.0) was chosen and what specific features were utilized for network construction.

Discussion

The discussion segment provides a thorough overview of the study's findings; however, there are some areas where clarity and precision can be improved:

a.     It would be important to discuss your findings in the context of previous studies on circRNA in muscle growth and development. How do your results align or differ from existing literature, especially in the context of geese compared to other species?

b.     It is equally essential for the authors to connect the identified hub genes and circRNA-miRNA-hub gene regulatory axes to the overall theme of the study, especially how they influence the proliferation and differentiation of myoblasts in Shitou geese. Explain the potential functional roles of these genes in the context of muscle development.

c.      When discussing the biological processes involving MSTN, lncRNA, MYL1, and ITGA8, make sure that the information is presented in a clear and accessible manner for readers who may not be familiar with the specific biological pathways. Define abbreviations where necessary.

d.    Clearly express the novel aspects of your study linking them to existing research. What distinctive contributions does your research make to the understanding of circRNA in the growth and development of goose muscle?

e.     Acknowledge the constraints or limitations earlier in the discussion to set the stage for interpreting the results. Also, specify the nature of the further experiments needed to address these limitations.

Conclusion

The conclusion section of the journal article provides a comprehensive summary of the study's findings. However, there are a few aspects that could be improved for clarity and completeness:

a)     Instead of using general terms like "novel insights" or "potentially participate," try to be more precise. Specify which circRNAs, microRNAs, mRNAs, or hub genes were identified, if possible. Provide brief and concrete examples or statistics to support your claims, making the conclusion more robust.

b)    Did you identify any limitations in your study, such as any constraints in the experimental design, sample size, or data analysis methods? If you did, then propose avenues for future research to address these limitations and expand on the current findings.

Comments on the Quality of English Language

I detected a few and pointed them out in my report.

Round 2

Reviewer 3 Report

Comments and Suggestions for Authors

You have done a great job in addressing all the suggestions I pointed out to you in the first round of the review. The manuscript no doubt looks good by my judgement.

Author Response

Dear reviewer,

Thank you for the comments.

It is your suggestion that makes our manuscript more perfect. Thank you again for your suggestion and confirmation.

Best wishes,

Wen Luo